# Direct Nanopore Sequencing of Human Cytomegalovirus Genomes from High-Viral-Load Clinical Samples

**DOI:** 10.3390/v15061248

**Published:** 2023-05-26

**Authors:** Kathy K. Li, Betty Lau, Nicolás M. Suárez, Salvatore Camiolo, Rory Gunson, Andrew J. Davison, Richard J. Orton

**Affiliations:** 1Medical Research Council, University of Glasgow Centre for Virus Research, Glasgow G61 1QH, UK; kathy.li@glasgow.ac.uk (K.K.L.); nicolas.suarez@glasgow.ac.uk (N.M.S.); andrew.davison@glasgow.ac.uk (A.J.D.); 2Regional Virus Laboratory, Belfast Health and Social Care Trust, Belfast BT12 6BA, UK; 3West of Scotland Specialist Virology Centre, NHS Greater Glasgow & Clyde, Glasgow G31 2ER, UK

**Keywords:** human cytomegalovirus, clinical sample, genome, nanopore sequencing, Illumina sequencing

## Abstract

Nanopore sequencing is becoming increasingly commonplace in clinical settings, particularly for diagnostic assessments and outbreak investigations, due to its portability, low cost, and ability to operate in near real-time. Although high sequencing error rates initially hampered the wider implementation of this technology, improvements have been made continually with each iteration of the sequencing hardware and base-calling software. Here, we assess the feasibility of using nanopore sequencing to determine the complete genomes of human cytomegalovirus (HCMV) in high-viral-load clinical samples without viral DNA enrichment, PCR amplification, or prior knowledge of the sequences. We utilised a hybrid bioinformatic approach that involved assembling the reads de novo, improving the consensus sequence by aligning reads to the best-matching genome from a collated set of published sequences, and polishing the improved consensus sequence. The final genomes from a urine sample and a lung sample, the former with an HCMV to human DNA load approximately 50 times greater than the latter, achieved 99.97 and 99.93% identity, respectively, to the benchmark genomes obtained independently by Illumina sequencing. Thus, we demonstrated that nanopore sequencing is capable of determining HCMV genomes directly from high-viral-load clinical samples with a high accuracy.

## 1. Introduction

Seropositivity to human cytomegalovirus (HCMV; species *Cytomegalovirus humanbeta5*) [1] ranges from 40% in developed countries to 100% in low- and middle-income countries (LMICs) [2,3]. Infection is usually asymptomatic [4], but in the absence of an appropriate immune response, may lead to severe morbidity and mortality, as exemplified in the transplant cohort [5] and the congenital setting [6]. Antiviral drugs are available but have myelotoxic and nephrotoxic profiles and are therefore best targeted to those diagnosed as being at highest risk. They also promote the generation of antiviral resistance mutants which are monitored on the basis of DNA sequence data in order to inform clinical management [7]. The identification of HCMV genes targeted by established antivirals and the resistance mutations associated with these genes is one application of HCMV sequence data, and this will continue to be the case as novel antivirals are developed in future. In keeping pace with relevant technologies over four decades, HCMV sequence data have also been pivotal in illuminating the structure, function, and evolution of HCMV genomes. The present ability to sequence HCMV genomes in large numbers also provides a powerful means of investigating the role of viral variation in disease outcome.

HCMV has a linear, double-stranded DNA genome of approximately 236 kbp [8,9]. The overall configuration is *ab*–U_L_–*b′a′c′*–U_S_–*ca*, in which unique regions U_L_ (193 kbp) and U_S_ (35 kbp) are flanked by inverted repeats *ab*/*b′a′* (1 kbp) and *a′c′*/*ca* (3 kbp). Early work involving restriction fragment length polymorphism analysis and Sanger sequencing revealed extensive genome variation among HCMV strains, including the existence of multiple discrete genotypes of a subset of hypervariable genes [10,11,12], the extensive rearrangement of these genotypes due to recombination during HCMV evolution [13], and the invariable loss of substantial genome segments and the accumulation of other mutations in highly passaged strains [14,15,16]. These studies were limited by the existing Sanger sequencing technology and typically focused on only one or a few genes. Subsequently, high-throughput, short-read sequencing was developed with a greater accuracy and at a lower cost [17], thus enabling the sequencing of large numbers of HCMV genomes [18,19,20,21,22] and facilitating the genotyping of hypervariable genes directly from read data [21,22,23]. These advances depended largely on data generated on the Illumina platform, which monitors the incorporation of labelled nucleotides during DNA copying in a hugely parallel fashion and has low error rates (<1% per base sequenced [24]). However, the correct determination of a novel HCMV genome from short-read data remains a challenge [23]. Repeat regions typically hinder the full reconstruction of an HCMV genome solely by de novo read assembly, and recombination, gene loss, and variability among HCMV strains result in suitable sequences usually not being available for reconstruction solely by aligning reads to a reference.

In recent years, there has been an acceleration in the development of high-throughput, long-read sequencing. The Oxford Nanopore Technologies (ONT) platform monitors disruptions in ionic current as DNA molecules pass individually through proteinaceous membrane pores in a highly parallel fashion. Read lengths on this platform can exceed 2 Mb [25,26], thus potentially enabling the entire HCMV genome to be sequenced in a single read. Nanopore sequencing also provides significant benefits for resolving repetitive regions and for investigating the occurrence of recombination [27]. In addition, low capital cost and portability have promoted the use of nanopore sequencing in the field for the rapid monitoring of viral outbreaks in LMICs [28] and in expediting near-real-time molecular epidemiological investigation during the COVID-19 pandemic [29,30]. Early limitations of the technology due to the high read error rate (30%) have been relieved by each iteration of the flow cell pores and base-calling algorithms [25,31,32,33], and recent upgrades employing a refined motor enzyme, improved flow cells, and enhanced software have further decreased the read error rate to <1% (https://nanoporetech.com/accuracy, accessed on 30 March 2023).

Nanopore sequencing has been applied to HCMV in several studies. In one study, nanopore data were generated from a laboratory strain (TB40/E) grown in cell culture and assembled de novo. The analysis was complicated by the existence of many variants in the viral stock used, but the accuracy of the consensus sequence (which was not published) in comparison with previously published sequences for this strain was reportedly high [34]. In a second study, nanopore data were generated from another laboratory strain (Merlin) grown in cell culture using bait-based target enrichment. This approach compromised one of the advantages of nanopore sequencing by significantly reducing DNA fragment size and therefore read length [35]. In a third study, the feasibility of nanopore sequencing was demonstrated in a clinical setting focused on antiviral resistance. This study was based on the nanopore sequencing of PCR amplicons from the two genes (UL54 and UL97) most likely to generate resistance mutations and demonstrated that nanopore data were concordant with those obtained by the usual approach of Sanger sequencing [36].

Here, we assessed the utility of direct nanopore sequencing (i.e., without viral DNA enrichment or PCR amplification) for determining complete HCMV genomes from high-viral-load clinical samples. Our finding that the nanopore sequences were highly accurate, although inferior in quality to the sequences determined on the Illumina platform, encourages a future investigation into the application of nanopore sequencing to research HCMV, including that in the clinical setting.

## 2. Materials and Methods

### 2.1. Sample Preparation and Characterisation

Two fully anonymised surplus diagnostic extracts from samples that had been prepared on the NucliSENS easyMAG (bioMerieux, Lyon, France) or Abbott m2000 (Abbott Laboratories, Chicago, IL, USA) platforms were provided by the West of Scotland Specialist Virology Centre. The samples had high ratios of HCMV to human DNA, as determined using qPCR assays for the HCMV UL97 gene (UL97F: ACCGTCTGCGCGAATGTTA; UL97R: TCGCAGATGAGCAGCTTCTG; UL97P: *VIC*-CACCCTGCTTTCCGAC-*MGB* (*VIC*, 2′-chloro-phenyl-1,4-dichloro-6-carboxyfluorescein; *MGB*, minor groove binder.)) and the human *FOXP2* gene (FOXP2_FWD: TCACTACTAACAATTCCTCCTCGACTAC; FOXP2_REV: GATGAGTTATTGGTGGTGATGCTT; FOXP2_PROBE: *VIC-*TCCTCCAACACTTCG-*MGB*). The amplification conditions were as follows: an initial step of 50 °C for 2 min, denaturation at 95 °C for 10 min, 40 cycles of 95 °C for 30 s, and annealing and elongation at 60 °C for 1 min. One sample had been extracted from urine and had viral and human DNA loads of 258,199 IU/µL and 323 genome copies/µL, respectively; 1 IU is approximately equivalent to 1 genome copy. The other sample had been extracted from lung tissue and had viral and human DNA loads of 172,259 IU/µL and 9636 genome copies/µL, respectively. The urine sample was therefore approximately 50 times more enriched in viral DNA than the lung sample.

The pipelines used to generate sequence data from the samples and assemble the HCMV genomes are summarised in Figure 1. The key steps in the Illumina pipeline (A1–A7) and the ONT pipeline (B1–B13) are summarised in the legend and referred to in the descriptions below.

### 2.2. Illumina Sequencing

DNA (0.29 µg and 0.58 µg from the urine and lung sample, respectively) was sheared to an average size of 500 bp using an LE220 sonicator (Covaris, Woburn, MA, USA). HCMV DNA enrichment using Agilent SureSelectXT (Agilent Technologies, Santa Clara, CA, USA) custom biotinylated RNA baits and a KAPA Biosystems sequencing library preparation kit were carried out as described previously [18,21,22] (A1). The sequencing libraries were indexed using ultrapure TruGrade oligonucleotides (Integrated DNA Technologies, Leuven, Belgium) and sequenced on a MiSeq instrument (Illumina, San Diego, CA, USA) (A2). The 4,652,420 and 5,699,896 paired-end reads of 150 nucleotides (nt) obtained from the urine and lung samples, respectively, formed the input Illumina data for analysis (A3).

### 2.3. Nanopore Sequencing

DNA (2.5 µg and 7.9 µg from the urine and lung samples, respectively) was processed into a sequencing library using an SQK-LSK109 ligation sequencing kit (ONT, Oxford, UK; B1). This involved treatment with an NEBNext FFPE DNA repair mix and an NEBNext Ultra II end repair/dA-tailing module (New England Biolabs, Ipswich, MA, USA), purification using Agencourt AMPure XP beads (Beckman Coulter, Brea, CA, USA), ligation of adapters using an NEBNext Quick ligation module (New England Biolabs), enrichment of large (>3 kb) fragments using long fragment buffer, and final purification in elution buffer. The libraries were loaded onto primed R9.4.1 flow cells (ONT) at concentrations of 17.7 and 37.4 ng/µL for the urine and lung samples, respectively, and sequencing was carried out for 72 h on a GridION instrument (ONT; B2). The reads were acquired using MinKNOW v4.3.11 (ONT; https://nanoporetech.com, accessed on 16 November 2021), and FAST5-formatted files were base-called using Guppy v4.0 (ONT; https://nanoporetech.com, accessed on 16 November 2021) in high accuracy mode with a minimum quality score of 7. The reads were assessed using MinIONQC v1.3.5 [37], and adapters were trimmed and chimeric reads with internal adapters removed using PoreChop v0.2.3 (https://github.com/rrwick/Porechop, accessed on 16 November 2021) (B3). The 10,616,519 and 8,731,192 reads obtained from the urine and lung samples, respectively, were then mapped to a collated set of 265 HCMV genomes representing all HCMV strains that had been sequenced at the time of analysis (April 2021; Appendix A) using Minimap2 v2.17 [38], and mapped reads were extracted using Samtools v1.9 [39]. The 37,334 and 11,841 HCMV-enriched reads obtained from the urine and lung samples, respectively, formed the input nanopore data for analysis (B4). Longer reads were obtained from the urine sample: for the urine and lung samples, respectively, mean read lengths were 1431 and 735 nt and maximum read lengths were 169,079 and 92,239 nt.

### 2.4. Genotyping Using Illumina and Nanopore Data

The genotypes of 13 hypervariable HCMV genes were determined from the input Illumina data using the genotyping module in the HCMV genome assembly program GRACy v0.4.4 [23] (A4) and from the input nanopore data using Minion_Genotyper v1.0 (https://github.com/salvocamiolo/minion_Genotyper/, accessed on 16 November 2021) (B5). Both programs enumerate reads containing conserved genotype-specific kmers for each hypervariable gene. These kmers were initially defined at one per genotype for 12 genes [21,22] and were then increased to include several per genotype for 13 genes [23].

### 2.5. Genome Determination Using Illumina Data

The Illumina input data were assembled using GRACy (A5). The draft genomes were assessed by trimming and quality-filtering the reads using Trim Galore v0.4.0 (https://www.bioinformatics.babraham.ac.uk/projects/trim_galore/, accessed on 10 November 2020) under the parameters --illumina --paired, aligning the filtered reads to the sequences using Bowtie 2 v2.3.1 [40] and Samtools v1.3 [38], inspecting the alignments visually using Tablet v1.21.02.08 [41] and making any necessary amendments (A6). Coverage statistics were generated using Tablet. These steps resulted in properly configured HCMV genomes (A7).

### 2.6. Genome Determination Using Nanopore Data

Draft genomes were determined using two parallel approaches, one involving de novo read assembly into contigs, and the other involving read alignment to a reference.

In the first approach (B6–B8), the nanopore input data for each sample were assembled into contigs using Canu v.1.9 [42] under the parameters genomeSize = 240,000 min ReadLength = 500 min OverlapLength = 50 -nanopore-raw (B6). The genome in the collated set of 265 genomes that best matched the longest contig was identified using BLASTN v2.4.0 (B7). The GenBank accession numbers of these genomes are KY490081.1 and KP745641.1 for the urine and lung genomes, respectively. The contigs were compared with the best-matching genome using MUMmer v4.0 and Mummerplot v3.5 [43] and rearranged manually via inspection of an alignment in AliView v1.27 [44] (B8). To accomplish this, a text editor was used to order and divide the contigs to ensure that the genome termini were at the ends of the sequences, to locate any gaps, and to represent the unique regions (U_L_ and U_S_) and inverted repeats (*ab*/*b’a’* and *a’c’*/*ac*) appropriately (unlike the situation with the Illumina genomes, the inverted repeats were not assumed to be identical). This approach generated an incomplete consensus genome.

In the second approach (B9–B10), the best-matching genome (from B7) was used as the reference for read mapping using Minimap2 (B9). BAM and mpileup files were generated from the alignment using Samtools v1.9 [39], and the mpileup files were parsed to output a consensus genome using VSensus (https://github.com/rjorton/VSensus, accessed on 10 November 2020) with the minimum coverage and minimum base quality parameters set at 20 and 0, respectively (B10). Coverage statistics were generated using weeSAM v1.5 (https://github.com/centre-for-virus-research/weeSAM/blob/master/weeSAM, accessed on 10 November 2020).

The genomes derived by the two approaches were aligned using MAFFT v7.310, and any gaps in the former were filled with data from the latter (B11). Finally, this draft genome was processed using Medaka v1.0.3 (https://github.com/nanoporetech/medaka, accessed on 10 November 2020), which has been trained to correct for common errors in ONT data (B12). These steps resulted in properly configured HCMV genomes that did not depend at any stage on the corresponding Illumina data (B13).

### 2.7. Analysis of Resistance Mutations

The coding sequences of genes UL97 and UL54 were analysed using a web-based mutation resistance analyser (MRA,; https://www.informatik.uni-ulm.de/ni/mitarbeiter/HKestler/mra/app/index.php?plugin=form, accessed on 16 June 2021) [45].

## 3. Results

Initial screening of the Illumina data to determine the number of HCMV strains present in the original samples was carried out using GRACy, and that of the nanopore data involved using a program devised for our study, Minion_Genotyper. Neither tool requires a genome but rather analyses the reads using a kmer-based approach, and each identified the same single genotype for each of the 13 hypervariable genes analysed (Table 1). This analysis confirmed the ability of Minion_Genotyper to genotype HCMV strains from nanopore data accurately and indicated that each sample contained a single strain. Subsequent analysis of the consensus genomes confirmed the genotypes. Consistent with the occurrence of recombination during HCMV evolution, the constellation of genotypes for each genome was unique in comparison with those reported previously [21,22].

Determination of the final Illumina genomes was accomplished using GRACy followed by manual amendments and ended with 86% of reads aligning to the urine genome (234,678 bp) at an average coverage depth of 2554 reads/nt and 66% of reads aligning to the lung genome (235,215 bp) at an average coverage depth of 2368 reads/nt. The final nanopore genomes were determined by a more complex series of steps. De novo read assembly resulted in a single contig (255,416 nt) corresponding to the entire HCMV genome and a partial repeat for the urine sample but in 20 contigs for the lung sample, with the four largest originating from the HCMV genome. The three largest of these contigs (37,585, 40,716, and 156,323 nt) represented the whole genome but for one gap of 1849 nt. Subsequent steps led to an intact nanopore genome for both samples, each with complete read coverage and determined independently of the corresponding Illumina genome. The analysis ended with 37,241 reads aligning to the urine genome at an average coverage depth of 203 reads/nt, and with 11,764 reads aligning to the lung genome at an average coverage depth of 26 reads/nt.

Comparisons between the Illumina and nanopore genomes revealed differences at 62 nt (0.03%) and 164 nt (0.07%) for the urine and lung samples, respectively (Appendix A). The great majority of differences (97 and 90%, respectively) were due to insertions or deletions (indels) associated with homopolymeric tracts that almost always caused frameshifting in the nanopore genomes when present in coding regions; this is a recognised issue with nanopore sequencing [33]. The Illumina version at the position of each difference was validated by a visual inspection of the read alignment. Moreover, comparisons with the corresponding regions in the collated set of 265 genomes supported the Illumina version more frequently than the nanopore version for the great majority of differences (95 and 96%, respectively). However, although Illumina data are known to represent the lengths of homopolymeric tracts more accurately than nanopore data [46], they tend to become heterogeneous with longer tracts due to natural genome variation, PCR error during sequencing library preparation, and lower coverage depth. Nonetheless, the Illumina genomes were more accurate than the nanopore genomes.

In clinical practice, the sequences of genes UL54 and UL97 are typically examined for resistance mutations when patients fail to respond to antiviral therapy. As a test of the utility of nanopore genomes for such investigations, the coding sequences of these genes were submitted to MRA, which hosts a curated database of published resistance mutations from a range of viruses including HCMV (Figure 2). The UL97 sequences of the Illumina and nanopore genomes from the urine sample were identical, whereas the UL54 sequence contained an erroneous deletion of 1 nt in the nanopore genome from the lung sample (Appendix A). MRA successfully registered the frameshift caused by this deletion. Differences associated with strain polymorphisms or mutations not associated with resistance were registered in both genes in both samples, but no mutations associated with drug resistance were found in either gene in either sample. As the clinical samples were fully anonymised, no information was available on whether the patients had received antiviral therapy. However, these findings illustrate the utility of analysing nanopore data using MRA.

## 4. Discussion

To our knowledge, our study is the first to use nanopore technology to determine complete HCMV genomes from clinical samples without involving cell culture, bait-based target enrichment, PCR amplification, incorporation of data for the same genome acquired on other platforms, or amendments based on prior knowledge of the target sequence. We sequenced two high-viral-load samples from urine and lung on the Illumina platform with bait-based target enrichment and independently on the ONT platform without enrichment. We used separate genotyping of the Illumina and nanopore data to demonstrate the presence of a single HCMV strain in each sample. The nanopore genomes were determined using a combination of two approaches, one involving de novo read assembly and the other involving read alignment to the best-matching HCMV genome, with the outputs being incorporated and polished to produce the final sequences. As anticipated from the approximately 50-fold greater ratio of viral to nonviral DNA in the urine sample, the urine genome was assembled more readily and at a greater read depth than the lung genome.

Sequence comparisons suggested that the differences between the Illumina and nanopore genomes were largely or entirely due to errors in the latter, as would be anticipated from the characteristics of the platforms used. Nonetheless, the nanopore genomes were highly accurate (99.97 and 99.93% for the urine and lung genomes, respectively), with the errors distributed thinly (on average, one approximately every 3.8 and 1.5 kbp, respectively) and located mostly in polynucleotide tracts in noncoding regions. Thus, although improvements in the ONT hardware and the software for analysing the data have steadily boosted the base accuracy to >99% (https://nanoporetech.com/accuracy, accessed on 30 March 2023), in our study they did not attain the level of accuracy achieved using the Illumina platform. For a completely accurate characterisation of HCMV strains on the ONT platform, supplementary data from another platform would be required. A further implication is that it would be inadvisable to rely on nanopore sequencing alone to determine the sequence of a novel large DNA virus for which no prior sequence information is available.

The main limitation of our study was that the two clinical samples were chosen for sequencing because they contained a high number of HCMV genomes and a relatively high ratio of HCMV to human DNA, thus meeting the requirements of nanopore sequencing. In a clinical setting, the mainstay sample type for post-transplant monitoring of HCMV reactivation or re-infection is blood, but HCMV genomes in blood are highly fragmented, thus obviating one of the chief advantages of nanopore sequencing; blood also contains high levels of host DNA, which can lead to low efficiency enrichment for HCMV genomes [47,48]. In contrast, urine is normally depleted of host cells and was shown in our study to be suitable for sequencing the HCMV genome with a high accuracy on the ONT platform. In instances of post-transplant HCMV disease refractory to treatment, urine is a readily obtained, non-invasive sample type. However, our protocol necessitates high HCMV to human DNA ratios, and for a broader application would require the incorporation of a target enrichment step that is capable of preserving large DNA molecules. The treatment of urine samples with DNase before DNA extraction may be advantageous in this regard if the majority of viral DNA is encapsidated.

Despite this limitation, nanopore technology potentially offers much to HCMV testing in clinical practice in the context of transplantation, particularly as the focus is usually on one or two genes, at which scale nanopore sequencing is almost as accurate as Illumina sequencing. In the UK, HCMV antiviral resistance testing is referred to specialist reference laboratories in which analysis is based on Sanger sequencing of the PCR-amplified UL97 and UL54 genes applied on a timescale that is not conducive to prompt clinical action. A nanopore protocol similar to our approach could provide a means of resistance testing on a shorter timescale. Recently, an Illumina sequencing pipeline was established that enables whole genome HCMV analysis for resistance mutations, which could facilitate resistance testing against novel drug targets [48]. Nanopore technology can also sequence whole HCMV genomes and, in addition, has a modest capital outlay and a small footprint. The continued development of online bioinformatic tools is also making nanopore technology increasingly accessible outside of the research context [36]. For lower viral load samples, PCR products covering antiviral genes could be sequenced using nanopore technology, potentially enabling a more rapid analysis. Furthermore, the application of nanopore technology could be envisaged for investigating congenital infection, where viral loads are typically high, in the order of that of the urine sample used in this study. As the management of congenitally infected infants is nuanced, with treatment reserved for a proportion of the 10% who are symptomatic with severe disease at birth [49], a more widely accessible sequencing system may prove advantageous, particularly if it transpires that there are links between certain HCMV strains, or mixtures of strains, and disease outcome. However, further work to establish thresholds for the detection of low-level variants as applied to Illumina sequence data will be required [47].

In conclusion, we were able to determine two HCMV genomes by direct nanopore sequencing from high-viral-load clinical samples at an accuracy that was close enough to that of Illumina sequencing to encourage further investigation. We await with interest additional improvements in the hardware and software used in this technology, as well as appropriate target enrichment techniques.

## Figures and Tables

**Figure 1 viruses-15-01248-f001:**
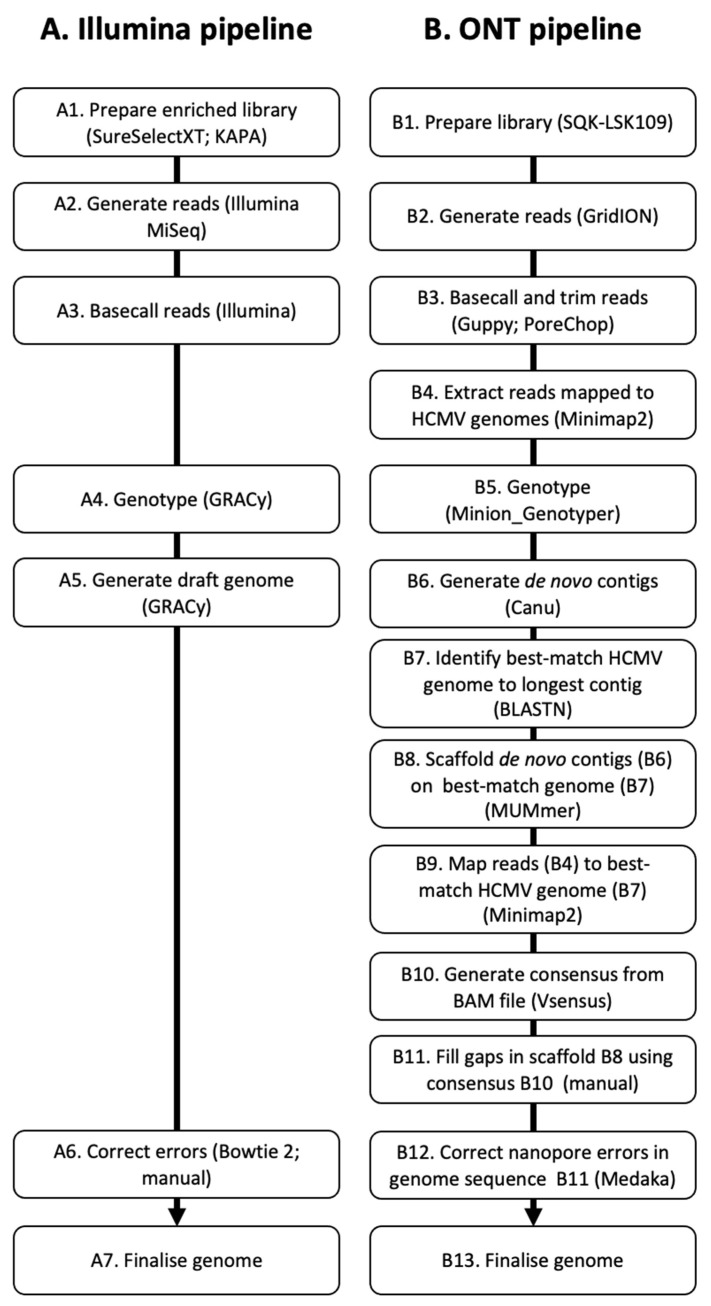
Sequencing and bioinformatics pipelines. The steps in (**A**) the Illumina pipeline and (**B**) the ONT pipeline consist of preparation of sequencing libraries (A1 and B1), generation of sequence reads (A2 and B2), extraction of read data (A3 and B3–B4), identification of hypervariable gene genotypes (A4 and B5), generation of draft genomes (A5 and B6), improvement of draft genomes (A6 and B7–B12), and finalisation of genomes (A7 and B13). The bioinformatic means by which the steps were achieved are given in parentheses. The steps are explained further in the text.

**Figure 2 viruses-15-01248-f002:**
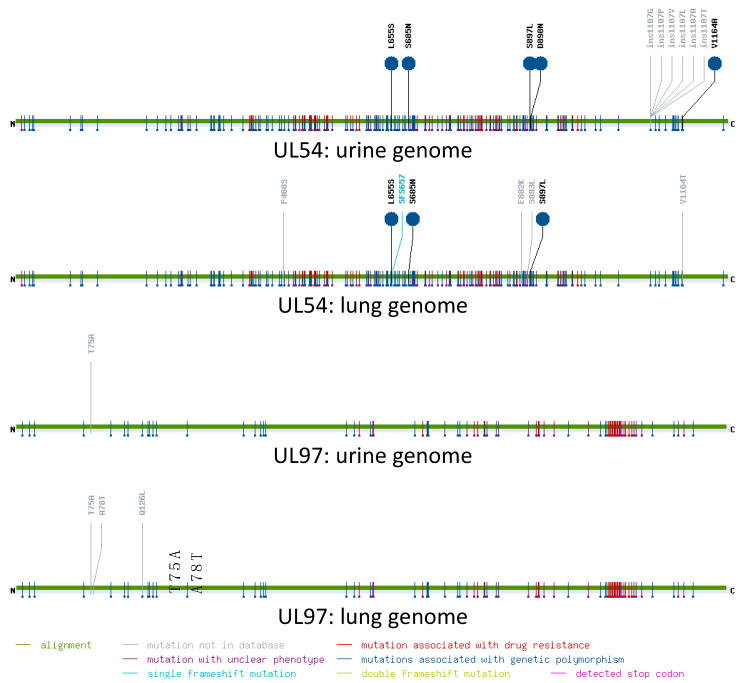
Graphical output from mutation resistance analyser (MRA) for the nanopore sequences of genes UL54 (3.7 kbp) and UL97 (2.1 kbp). In each panel, the sequence alignment is represented by a line extending from the 5′ end encoding the N terminus of the protein (N) to the 3′ end encoding the C terminus (C). Mutations are shown by the flags coloured according to the key at the foot. Mutations flagged above the line are present in the sequences analysed, and mutations below the line are those in the curated database. The coordinates relate to encoded amino acid residues.

**Table 1 viruses-15-01248-t001:** Genotypes of 13 hypervariable HCMV genes in the urine and lung samples.

Gene	Urine	Lung
RL5A	G1	G2
RL6	G6	G4
RL12	G7	G1B
RL13	G7	G1
UL1	G7	G1
UL9	G1	G4
UL11	G1	G1
UL20	G6	G6
UL73	G4D	G1
UL74	G5	G1A
UL120	G1A	G4B
UL146	G2	G10
UL139	G1A	G2

## Data Availability

The Illumina genomes for the lung and urine samples have been deposited in GenBank (accessions OQ466311 and OQ466312, respectively). The nanopore genomes for these samples are available on Zenodo (https://doi.org/10.5281/zenodo.7744466 accessed on 30 March 2023). The HCMV reads supporting these genomes have been deposited in the NCBI Sequence Read Archive (SRA) [50] under BioProject number PRJNA945420.

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
