# Peer review of "Direct Nanopore Sequencing of Human Cytomegalovirus Genomes from High-Viral-Load Clinical Samples"

_viruses, 2023, doi:10.3390/v15061248_

Round 1
Reviewer 1 Report
Comments and Suggestions for Authors
See Comments and Suggestions attached with file "Reviewer comments on Li et al (viruses-2382873).pdf"

Reviewer 2 Report
Comments and Suggestions for Authors
In this submission, Li and coworkers present state-of-the-art evaluation of the feasibility of using new long-read technology from Nanopore for determining full-length human cytomegalovirus genome sequences from clinical specimens, in the absence of virus propagation in cultured cells. The evaluation was done in comparison to high-throughput Illumina sequencing of randomly-fragmented DNA. Long-read technology offers the ability to sequence single molecules of DNA through regions of repeats and diverse sequence rearrangements, which is of great value when examining the complex forms of repetitive and rearranged sequences present in herpesvirus genomes. The authors found the Nanopore-generated data to be almost as accurate as data obtained by Illumina sequencing. Importantly, they provide a clearly described computational pipeline for assembling clean sequences from the Nanopore data. This will be of use to others.
The paper is very cleanly prepared, and clearly written. My comments are minor.
1. Please discuss the potential value of treating the biological specimen with DNAse first, to enable enrichment for encapsidated viral DNA that would likely provide even longer reads.
2. Define “LMIC” (line 62)
3. The description of Fig. 2 can be improved. Information such as “Urine genome UL54” would be better as part of the diagram, rather than in the legend. The legend is not clear with respect to the nature of information shown above vs. below each horizontal line. A scale would be helpful.
Reviewer 3 Report
Comments and Suggestions for Authors
Summary: Advanced sequencing is becoming more applicable to viral characterization. The objective of this manuscript is to assess the ability of the ONT MinION Platform for direct sequencing and characterization of human cytomegalovirus from clinical samples. The manuscript is well written. The comparison to the Illumina-based sequencing is appropriate. The manuscript would have benefitted from additional samples, but this is addressed in the manuscript. The most significant recommendation is to elaborate on the materials and methods for the development of the Minion_Genotype program. For example, what defines the genotypes and what supports the genotypes. This reviewer assumes that similar information as to what was presented in reference 19, but with multiple motifs in this newest manuscript, additional information on this program would strengthen the manuscript.
Minor Revisions:
1. For the homopolymer inaccuracy of MinION sequencing, it is recommended to cite this manuscript (or something similar), in which this known issue is reported.
a. Rang FJ, Kloosterman WP, de Ridder J. From squiggle to basepair: computational approaches for improving nanopore sequencing read accuracy. Genome Biol. 2018 Jul 13;19(1):90. doi: 10.1186/s13059-018-1462-9. PMID: 30005597; PMCID: PMC6045860.
